# Strengthening systems to provide long-acting reversible contraceptives (LARCs) in public sector health facilities in Uganda and Zambia: Program results and learnings

Aniset Kamanga[1☯], Micheal Lyazi[2☯], Margaret L. Prust[3], Naomi Medina-Jaudes[3], Lupenshyo Ngosa[1], Margaret Nalwabwe[2], Martha Ndhlovu[1], Dynes Kaluba[4], Angel Mwiche[4], Richard Mugahi[5], Joy Batusa[2], Morrison Zulu[1], Andrew Musoke[2], Hilda Shakwele[1], Caitlin Glover[3], Emma Aldrich[3]*

1 Clinton Health Access Initiative, Inc., Lusaka, Zambia, 2 Clinton Health Access Initiative, Inc., Kampala, Uganda, 3 Clinton Health Access Initiative, Inc., Boston, MA, United States of America, 4 Department of Public Health, Zambia Ministry of Health, Ndeke House, Lusaka, Zambia, 5 Uganda Ministry of Health, Reproductive and Infant Health, Kampala, Uganda

☯ These authors contributed equally to this work.
* ealdrich@clintonhealthaccess.org

## Abstract

### Introduction

In Uganda and Zambia, both supply- and demand-side factors hamper availability of long-acting reversible contraceptives (LARCs), including implants and intrauterine devices (IUDs), at public sector facilities. This paper discusses results of a program aimed at increasing access to and uptake of LARC services in public sector facilities through capacity building of government health workers, strengthening government supply chains, and client mobilization.

### Methods

From 2018–2021, the Ministries of Health (MOHs) in Uganda and Zambia and Clinton Health Access Initiative (CHAI) worked to increase readiness to provide LARC services within 51 focal facilities in Uganda and 85 focal facilities in Zambia. Annual facility assessments of LARC-related resources were conducted and routine service delivery data were monitored.

### Results

At baseline, few focal facilities had supplies and skilled staff to provide LARC services. At endline, over 90% of focal facilities in both countries had a provider trained to provide both implants and IUDs and 55% had the commodities and equipment needed for implant provision. In Uganda and Zambia, respectively, 65% and 38% of focal facilities had commodities and equipment for IUD provision at endline. Both programs observed significant increases in the number of implants provided at focal facilities; in Uganda implant volumes increased

**Data Availability Statement:** All relevant data are within the paper and its Supporting Information files.

**Funding:** This work was made possible with financial support from The ELMA Foundation and an anonymous donor (grants received and managed by HS and AM). The views expressed in this report are the opinions of the authors, and do not necessarily reflect the official policies of the funders. The funders had no role in study design, data collection and analysis, decision to publish, or preparation of the manuscript.

**Competing interests:** The authors have declared that no competing interests exist.

five-fold from 4,560 at baseline to 23,463 at endline, and in Zambia implant volumes increased nearly four-fold from 1,884 at baseline to 7,394 at endline. Uganda did not observe growth in IUD volumes, whereas Zambia observed significantly increased IUD service volumes from 251 at baseline to 3,866 at endline.

## Conclusions

Public sector facilities can be rapidly and sustainably capacitated to provide LARCs when both catalytic and systems strengthening interventions are deployed for health worker capacity building, supply chain management, and community mobilization to ensure client flow. Investments should be intentionally sequenced and coordinated to generate a virtuous cycle that enables continued LARC service provision.

## Introduction

In Uganda and Zambia, high numbers of unintended pregnancies contribute to high rates of morbidity and mortality due to unsafe abortion and lack of access to routine and emergency obstetric care. The governments of Uganda and Zambia are committed to increasing access to and use of modern family planning (FP) to reduce unintended pregnancies and maternal morbidity and mortality. For example, the Zambian National Health Strategic Plan of 2022–2026 commits to increase the proportion of women of reproductive age (aged 15–49 years) who have their need for family planning satisfied with modern methods from 68.5% in 2018 to 80% 2026 [1], and in Uganda, the government seeks to increase the modern contraceptive (mCPR), for all women, from 30.4% in 2020 to 39.6% by 2025 and reduce the unmet need for modern contraception from 17% in 2020 to 15% by 2025 [2]. Unintended pregnancies are pregnancies that are mistimed, unplanned or unwanted at the time on conception. Efforts have been made to scale access to long-acting reversible contraceptive (LARC) methods, including implants and intrauterine devices (IUDs), because they are more effective [3] and have higher continuation rates compared to short-acting methods, which require end-users to frequently re-supply and are subject to user error [4]. In addition, LARC methods are more cost-effective on a cost per couple years of protection (CYP) basis, an important consideration as countries move to increase domestic financing for FP product procurement and service delivery [5,6].

Supply- and demand-side factors, however, continue to hamper uptake of LARC services in Uganda and Zambia, especially in the public sector, where most modern contraceptive users receive services [7,8]. Public sector supply-side gaps include insufficient numbers of trained health workers skilled to provide LARC services, persistent gaps in LARC commodities at service delivery points, and lack of essential equipment, medicines and supplies to provide LARCs. At the same time, demand-side issues related to family planning include lack of awareness of LARC benefits and pervasive myths and misconceptions about implants and IUDs [9,10]. As a result, only 17.3% of all contraceptive users in Uganda use implants and 4.1% use IUDs, with overall method share skewed toward short-term contraceptives, such as injectables (51.3%) [11]. Similarly, in Zambia, 17.9% of all contraceptive users use implants and only 1.5% use IUDs, with method share also skewed towards short-term methods, notably injectables (52.8%) [12].

The Zambian and Uganda governments are working to address these issues and expand access to family planning, including LARCs. In Zambia, the government pledged to invest $12

million in family planning programs in 2023, with yearly increases of 30% after that [13]. In Uganda, the government has committed to ring fence 50% of the domestic reproductive health commodities budget for procurement, storage and distribution of FP commodities from by 2025 [14]. Alongside the efforts of the government, international non-governmental organizations (INGOs) have implemented LARC service delivery and demand generation interventions in Uganda and Zambia, circumventing capacity issues in the public sector by deploying donor-funded mobile clinical outreach teams or seconding dedicated family planning providers to the public sector, and by building the capacity of private-sector franchised or affiliated clinics [15]. Neither implementation experiences nor evaluations of the effectiveness of public sector strengthening programs have been widely documented. This paper presents the results of a program that aimed to increase access to and uptake of LARC services within public sector facilities in Uganda and Zambia by building capacity of government-employed health workers while strengthening government systems for service delivery and client mobilization.

Furthermore, the ways in which supply- and demand-related dynamics create a mutually reinforcing cycle, either enabling or inhibiting LARC uptake, have not been well explored in published literature. While low levels of LARC uptake are often attributed to gaps in availability of trained health workers, relevant supplies, or persistent myths and misconceptions, it is often the case that there are relationships between supply and demand factors that contribute to either vicious or virtuous cycles. **Fig 1** depicts a virtuous cycle for LARC introduction as developed and used by this program. Anecdotally, it's been observed that health worker skills may deteriorate post-training when they do not consistently perform the skill immediately following training due to commodity stock outs or insufficient demand and subsequently, client flow. In the absence of frequent opportunities to practice skills and provide the service, health worker competency and confidence to provide the service declines. Because health workers are less likely to provide comprehensive information on methods that they are uncomfortable providing, they're likely to promote or focus on other contraceptive options, missing opportunities to correct misperceptions and share information on the benefits of LARCs during FP counselling sessions, even if LARC options are more likely to meet clients' reproductive goals. Having not had their concerns about LARCs addressed, clients go on to select alternative FP options, including popular injectables, and are likely to promote use of injectables or other short-term methods amongst their peer networks. Health workers then interpret the absence of feedback from satisfied LARC users as a signal that clients prefer short-term methods. Given these dynamics, the return on investment on singular, mass trainings which are deployed without consideration for the need to strengthen supply chain and community

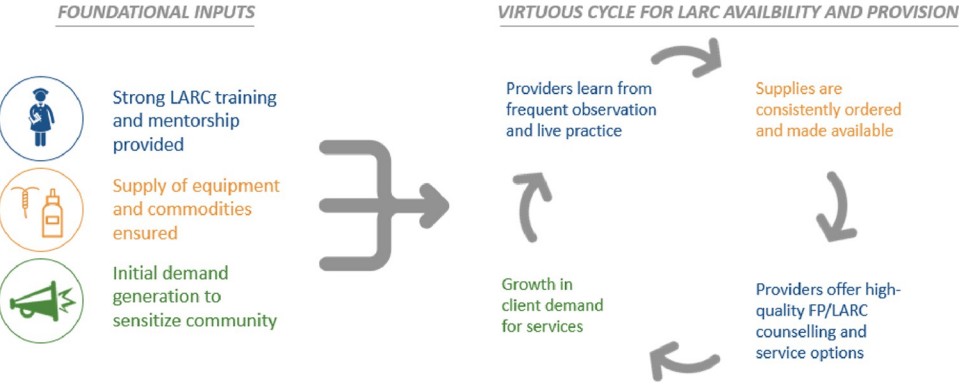

**Fig 1. Virtuous cycle of LARC availability and provision.**

mobilization systems in tandem is likely to be limited. Programs that employ intentional sequencing of multiple and at times synchronous interventions to disrupt a vicious cycle and establish a new virtuous cycle of supply and demand for LARC uptake have not been widely documented. This paper presents results of a program which addressed immediate gaps preventing service readiness at public sector facilities, while also aiming to establish a virtuous cycle of supply and demand for LARC services, for sustainability and optimal return on investment over time.

## Methods

### Program approach

From 2018–2021, the Ministries of Health (MOH) in Uganda and Zambia and Clinton Health Access Initiative, Inc. (CHAI), with funding from The ELMA Foundation and an anonymous donor, worked in partnership to implement an integrated Sexual, Reproductive, Maternal, and Newborn Health (SRMNH) program in selected sub-national geographies in Uganda and Zambia [16]. The program was designed by CHAI and the MOHs, building on previous experience of both countries and with careful consideration and adaptation for the local context and opportunities. CHAI developed a standardized mentorship framework and implementation package to assist focal geographies to address systems and facility-level barriers to SRH access, provision, and uptake. This assisted in addressing the gaps in LARC service provision. CHAI leveraged the pool of family planning government trainers and mentors to build health care worker capacity and drive acknowledgement amongst community traditional leadership of the need to scale up awareness of the benefits of family planning. The main program activities were implemented over a three-year period from 2018 through 2020, whereas the program focused on transitioning technical assistance and ensuring the sustainability of interventions in 2021.

One component of the programs in Uganda and Zambia was to demonstrate how technical assistance partners, local government units and health managers can work together to rapidly improve the availability of LARC methods and services in eligible public sector facilities. The program activities relating to LARCs that were implemented in Uganda and Zambia by CHAI and the government are described in detail in **S1 Table**. Though the activities were implemented with government collaboration, these activities were beyond and in addition to the routine service support activities performed by the government. These activities were synchronous and rationally paced and aimed to create a virtuous cycle of supply and demand for LARCs while strengthening government systems for sustainable delivery of LARC services and to address systemic barriers to LARC provision and use. Key interventions were implemented in relation to the three foundational input areas described in the virtuous cycle schematic (health worker capacity building, availability of equipment and commodities, and demand generation) as well as in support of strengthening management and information systems to improve availability of routine data to aid decision-making. Interventions were largely implemented by and through existing sub-national government structures and systems, with the aims of ensuring government ownership and sustainable impact over time and facilitating seamless withdrawal of technical assistance at the end of the program. In both Uganda and Zambia, leveraging program interventions, government stakeholders elected to add training, mentorship, peer educator outreach, and supply chain quality improvement activities to annual subnational workplans and, where possible, domestic government budgets to institutionalize LARC service delivery as part of public sector standard operations. Initial training and catalytic supply chain interventions, as well as targeted community mobilization were timed and deployed to increase client flow immediately ahead of health worker trainings and

to ensure that LARC-related supplies were available for service delivery post-training. It was particularly important to ensure: LARC products were procured, distributed and available at facilities immediately following training, to ensure health workers could put skills to use immediately, minimizing risk of skills and confidence atrophy; LARC products were not procured too far in advance of training, to avoid running down shelf-life and risk of expiry and wastage; clients were mobilized once training dates were confirmed, to ensure sufficient client flow for training practicum and routine service delivery following training. On-the-job clinical mentoring, supply chain management capacity building, and routine community mobilization activities were then deployed synchronously to support integration of clinical skills into routine service delivery and support facilities to be able to consistently meet increasing demand for services over time, without interruption.

In both countries, the following LARC products were routinely procured through public sector procurement channels during the program period: single-rod etonogestrel implant (Implanon NXT); dual-rod levonorgestrel implant (Jadelle); and non-hormonal (copper) IUD products. Training and mentorship was offered to cadres eligible to provide LARC services in each country, with a focus on nurses, midwives, and clinical officers. Facilities with trained health workers received on-the-job mentoring. Mentorship helped to institutionalize skills within facilities and benefitted multiple health workers beyond those that were trained formally by the program. While mentoring may have helped to cascade skills and mitigate impacts of staff transfers, provider attrition was a reality the program had to contend with. The program coordinated with local leadership to rationalize staff transfers and ensure coverage of trained health workers at program sites. Trained health workers were encouraged to provide on-the-job training to other health workers in their facilities to ensure the availability of services in the event of attrition.

## Setting

In Uganda, the program provided support to 51 LARC-eligible government-owned health facilities in six districts, including three hospitals and 48 health centers. These 51 facilities serve an estimated 2,025,000 people in the Kagadi, Kakumiro, Kassanda, Kibaale, Mityana, and Mubende districts. Facilities designated as a health center level three (HCIII) or higher with a midwife are eligible to provide LARC services in Uganda per the 2016 Uganda Clinical Guidelines [17].

In Zambia, the program provided support to 85 LARC-eligible government-owned health facilities in all 12 districts in Northern Province. Together, these facilities serve an estimated 1,520,004 people in Northern Province. All facilities designated as a health center or higher are eligible to provide LARCs in Zambia as per the 2006 Zambia Family Planning Guidelines and Protocols [18].

These areas of Zambia and Uganda were selected by the government as the focal area for this work because they were some of the most underperforming areas of the country on SRMNH-related outcomes, Northern Province, Zambia is a very rural and hard-to-reach areas with no other maternal and newborn health programs active in the province at the time that this program was launched. The selected districts in Uganda are within several hours drive from the capital of Uganda, Kampala, but nonetheless include areas that are hard-to-reach due to poor road networks.

## Facility assessments

Facility assessments were conducted in Uganda and Zambia to determine which facilities had the resources and capacity to provide LARCs and other SRMNH services. The facility

**Table 1. Key LARC-related indicators in Uganda and Zambia.**

| Indicator | Definition |
|---|---|
| *Service readiness* | |
| Percent of facilities that have at least one trained provider for implants / IUDs | Proportion of LARC-eligible program facilities that had at least one provider trained to provide implants / IUDs |
| Percent of facilities that have at least one mentored provider for implants / IUDs | Proportion of LARC-eligible program facilities that had at least one provider mentored to provide implants / IUDs |
| Percent of facilities that have implant / IUD supply | Proportion of LARC-eligible program facilities that had at least one implant / IUD available |
| Percent of facilities with tracer commodities/ equipment for implants | Proportion of LARC-eligible program facilities that had tracer commodities/equipment for implants, including: implant, sterile gloves, betadine, cotton balls, lidocaine, and sterile drapes |
| Percent of facilities with tracer commodities/ equipment for IUDs | Proportion of LARC-eligible program-facilities that had tracer commodities/equipment for IUDs, including: IUD, sterile gloves, sterile scissors, speculum, tenaculum, and sponge holding forceps |
| *Service provision* | |
| Percent of facilities providing implants / IUDs | Proportion of LARC-eligible program facilities that provided at least 1 implant / IUD in the last 90 days |
| Implant / IUD service volumes | Number of implants / IUDs provided at LARC-eligible program facilities |
| Percent of CYPs generated by implants / IUDs | Proportion of couple years of protection generated by implants / IUDs at LARC-eligible program facilities |

*Slash marks (/) indicate two distinct indicators: One for implants and one for IUDs.

assessments were conducted by program staff or contractors using tools developed by the program to track the intended program indicators (**Table 1**) on family planning and other topics. The baseline assessment was conducted in July and August 2018 for Uganda and Zambia, respectively. Beginning in 2019, facility assessments were conducted quarterly through the end of 2020. For both countries, a data collection tool was adapted from the WHO Service Availability and Readiness Assessment (SARA) tool. In Zambia, the draft form for the MOH Service Quality Assessment (SQA) tool also informed the development of the facility assessment data collection tool. The facility assessments were approved by ethical review committees in both countries in July 2018, including the ERES Converge Institutional Review Board (IRB) in Zambia (reference number 2018-Jul-017) the AIDS Support Organization IRB in Uganda (reference number 026/18-UG-REC-009). The IRBs determined that this work did not engage human subjects and that patient consent was not required.

Data were collected in person by CHAI and MOH staff through observations in facilities, administrative record review, and by asking facility leadership questions about service delivery and resource availability. The data collection tools were administered using tablets and SurveyCTO for electronic data collection. Data were collected by phone in 2020, when Uganda and Zambia experienced lockdowns and facility visits were not feasible due to the COVID-19 pandemic.

### Routine service delivery data

To track the provision of family planning services at program supported sites, data was extracted from the national health information management system (HMIS), which is hosted on the DHIS2 platform in both countries. CHAI and MOH staff had access to aggregated monthly data. Extreme outliers were observed in rare cases and were followed-up to enable correction of data entry errors. In both countries, INGOs operate mobile clinical outreach teams which are hosted by lower-level public sector health facilities and in some cases, rural communities without a health center, and which provide free family planning services to

public sector clients. In both countries, LARC services provided by INGO-employed clinicians via mobile outreach events are recorded in host facility HMIS forms and are reflected in DHIS2 as services provided by that facility.

This data is available in aggregate form without any individualized patient information, and as a result, this work was determined by CHAI's internal Scientific and Ethical Review Committee to not engage human subjects or require patient consent.

## Key indicators

In relation to LARCs, several key indicators were agreed upon at the outset of the program by program and government staff as a foundation to align all stakeholders around strategic goals and measures of success (**Table 1**). The facility assessment data were used to track service readiness. We define readiness as the presence of the necessary commodities, equipment, and trained staff to provide a specific service. Specifically, for IUDs, the following commodities had to be available and functional on the day of the assessment: IUD, sterile gloves, sterile scissors, speculum, tenaculum, and sponge holding forceps. For implants, facilities were required to have the following commodities available and functional on the day of the assessment: implant, sterile gloves, betadine, cotton balls, lidocaine, and sterile drapes. Facilities were also required to have at least one staff member ever trained in each LARC method to be considered ready to provide the service. In addition, the program tracked the proportion of facilities that had a staff member mentored on each LARC service because mentorship supports integration of skills into clinical practice and helps reinforce skills and confidence over time. Staff were considered trained if they had ever received training in IUD and implant provision, and mentored if they had received at least one targeted, structured mentorship session within the last 12 months. Only staff currently working at the facility and who were not on leave for more than 90 days were counted.

Service provision indicators were monitored from the national HMIS in both countries. Data were available on the number of IUDs and implants reported as provided at each program facility in each month and the total number of IUDs and implants reported as provided over time was also tracked. The program also tracked the percentage of facilities that reported providing at least one implant or IUD per quarter as a measure of which sites were actively providing services. In addition, the program tracked the CYPs generated by LARCs and other modern methods [6]. Specifically, we assessed the percentage of CYPs generated by implants and IUDs out of total CYPs generated by oral contraceptive pills, injectables, implants, and IUDs provided.

## Data analysis and program data use

Data in both Uganda and Zambia were monitored using country-specific dashboards that were developed and maintained in GoogleSheets for the duration of the program. The dashboards were linked to the SurveyCTO data collection tools so that a data tab was populated automatically as completed forms were submitted. This enabled monitoring of data as it was being collected and entered into the system, and led to timely correction of the few outliers and errors that were committed during the process. Analysis tabs were created to visualize and summarize the data on which facilities met LARC service readiness requirements and which facilities were providing LARCs. Final additional analyses of data were done using Stata 14. The datasets used for analysis are available with this paper in **S1** and **S2 Files**. Significance testing to compare the baseline and endline values using matched facility-level data was done using McNemar's chi-squared tests for binary outcomes and Wilcoxon signed-rank test for continuous outcomes.

## Results

In Uganda, a high proportion of focal facilities were already reporting routine provision of LARC services at baseline; 88% and 73% of facilities had provided implants and IUDs, respectively, within the last 90 days (**Table 2**). In Zambia, a lower percentage of facilities were routinely providing LARC services at the start of the program, with only 54% of facilities providing implants and 15% of facilities providing IUDs within the last 90 days at baseline. With the notable exception of IUD service provision in Zambia, most focal facilities were reporting providing LARC services on a routine basis (at least once within a 90-day period) prior to the introduction of the program in 2018, though due to the setup of HMIS reporting forms and DHIS2 reporting categories, it is not possible to ascertain the proportion of LARC services directly provided by government health facilities and health workers versus via INGO mobile clinical outreach teams. In contrast to relatively high baseline coverage of services, in

**Table 2. For key LARC program indicators in Zambia and Uganda.**

| | Zambia | | | | Uganda | | | |
|---|---|---|---|---|---|---|---|---|
| Data | Baseline | Midline | Endline | p-value[1] | Baseline | Midline | Endline | p-value[1] |
| **Service readiness[2]** | | | | | | | | |
| % of program facilities with a trained provider in implant | 60% (51/85) | 74% (63/85) | 92% (78/85) | <0.001 | 84% (43/51) | 94% (48/51) | 98% (50/51) | 0.008 |
| % of program facilities with a mentored provider in implant | 9% (8/85) | 87% (74/85) | 99% (84/85) | <0.001 | 53% (27/51) | 94% (48/51) | 98% (50/51) | <0.001 |
| % of program facilities with a trained provider in IUD | 54% (46/85) | 76% (65/85) | 92% (78/85) | <0.001 | 45% (23/51) | 100% (51/51) | 96% (49/51) | <0.001 |
| % of program facilities with a mentored provider in IUD | 4% (3/85) | 84% (71/85) | 98% (83/85) | <0.001 | 31% (16/51) | 94% (48/51) | 98% (50/51) | <0.001 |
| % of program facilities with implant stock | 86% (73/85) | 73% (62/85) | 62% (53/85) | <0.001 | 84% (43/51) | 94% (48/51) | 98% (50/51) | 0.008 |
| % of program facilities with IUD stock | 38% (32/85) | 62% (53/85) | 39% (33/85) | 0.862 | 61% (31/51) | 92% (47/51) | 94% (48/51) | <0.001 |
| % of program facilities with tracer commodities/equipment for implants[3] | 2% (2/85) | 28% (24/85) | 55% (47/85) | <0.001 | 4% (2/51) | 37% (19/51) | 55% (28/51) | <0.001 |
| % of program facilities with tracer commodities/equipment for IUDs[4] | 12% (10/85) | 29% (28/85) | 38% (32/85) | <0.001 | 10% (5/51) | 71% (36/51) | 65% (33/51) | <0.001 |
| **Service provision[5]** | | | | | | | | |
| % of program facilities that provided at least one implant in last 90 days[6] | 54% (46/85) | 60% (51/85) | 68% (58/85) | 0.023 | 88% (45/51) | 92% (47/51) | 96% (49/51) | 0.103 |
| % of program facilities that provided at least one IUD in last 90 days[6] | 15% (13/85) | 39% (33/85) | 32% (27/85) | 0.004 | 73% (37/51) | 73% (37/51) | 92% (47/51) | 0.012 |
| Number of implants provided at program facilities[7] | 1,884 | 7,184 | 7,394 | <0.001 | 4,560 | 10,346 | 23,463 | <0.001 |
| Number of IUDs provided at program facilities[7] | 251 | 4,384 | 3,866 | <0.001 | 2,981 | 1,686 | 2,199 | 0.015 |
| % of CYPs generated by implants[7] | 38% | 47% | 53% | <0.001 | 48% | 72% | 82% | <0.001 |
| % of CYPs generated by IUDs[7] | 6% | 35% | 33% | <0.001 | 43% | 16% | 12% | 0.101 |

P values are based on a comparison of baseline and endline values using matched facility-level with McNemar's chi-squared tests for binary outcomes and Wilcoxon signed-rank test for continuous outcomes.

All service readiness data points are from the facility assessment data. For Uganda, baseline was July 2018 and for Zambia baseline was September 2018. For both countries, midline was December 2019 and endline was December 2020.

Implant tracer commodities included: Implant, sterile gloves, betadine, cotton balls, lidocaine, and sterile drapes

IUD tracer commodities included: IUD, sterile gloves, sterile scissors, speculum, tenaculum, and sponge holding forceps

All service provision data points are from the national HMIS data.

For both countries, baseline was April to June 2018, midline was October to December 2019, and endline was October to December 2020.

For both countries, baseline was January to June 2018, midline was July to December 2019, and endline was July to December 2020.

both countries, the baseline assessment found notable gaps in availability of the supplies and equipment needed to provide implant and IUD services at these facilities (described below). The program first assessed service readiness and identified the gaps that might inhibit consistent availability or quality of services. Following baseline assessments, the program tracked service provision and service volumes to monitor trends in provision of services over time, with the expectation that both availability and provision of services would increase as more facilities were consistently equipped with required resources, as health worker confidence increased with mounting clinical experience, and as positive user experiences stimulated demand in communities.

## Service readiness

At the start of the program in 2018, although most focal facilities in Uganda were providing both implants and IUDs routinely and most focal facilities in Zambia routinely provided implants, few focal facilities in either Uganda or Zambia had all human resources and supplies needed to provide implant and IUD services consistently without interruption (**Table 2**). In Uganda, the proportion of facilities that had a provider trained to provide implants increased from 84% at baseline to 98% at endline. This was coupled with an improvement in the availability of the required commodities and equipment to provide implants. Regarding service readiness for IUDs, 45% of LARC-eligible focal facilities had a provider trained to provide IUDs at the beginning of the program and this increased to 96% by the end of 2020. In the same period, commodity and equipment availability for providing IUDs increased substantially.

Similar trends were reflected in the Zambia program. The proportion of LARC-eligible focal facilities that had a provider trained to provide implants increased from 60% at baseline to 92% at endline. This was coupled with an improvement in the availability of all the required commodities and equipment to provide implants from 2% at baseline to 55% at endline. Regarding service readiness to provide IUDs, 54% of LARC-eligible focal facilities had a provider trained to provide IUDs at the beginning of the program and this increased to 92% by the end of 2020. In the same period, commodity and equipment availability for providing IUDs increased, though not as dramatically as in Uganda.

In both countries, the increase in the number of facilities that had a trained provider were smaller than the increase in facilities with all necessary commodities and equipment to provide LARCs. Importantly, the indicator on facilities with one staff member trained does not capture measurement of other improvements in health worker capacity achieved through having skills refreshed or having multiple staff members trained in a service area. In addition to training, on-the-job, post-training mentoring was also provided by government clinical mentors to health workers to support integration of skills into routine service delivery and drive provider confidence and quality of care. In Uganda, at endline, 98% of focal facilities had providers mentored to provide implants and IUDs, compared to only 53% and 31% respectively at baseline. In Zambia, 99% and 98% of focal facilities had providers mentored to improve implants and IUDs respectively at endline, compared to 9% and 4% at baseline.

## Service provision

An increase in the number of implant and IUD services provided in both countries from 2018 to 2020 (**Table 2**) reflected the increase in number of facilities that met the minimum staffing and commodity requirements to provide LARCs in Uganda and Zambia, and that were also provided mentorship to support integration of skills into daily service delivery post-training.

As stated previously, coverage of service provision of LARCs in Uganda was relatively high at baseline; 88% of LARC-eligible focal facilities had provided at least one implant and 73%

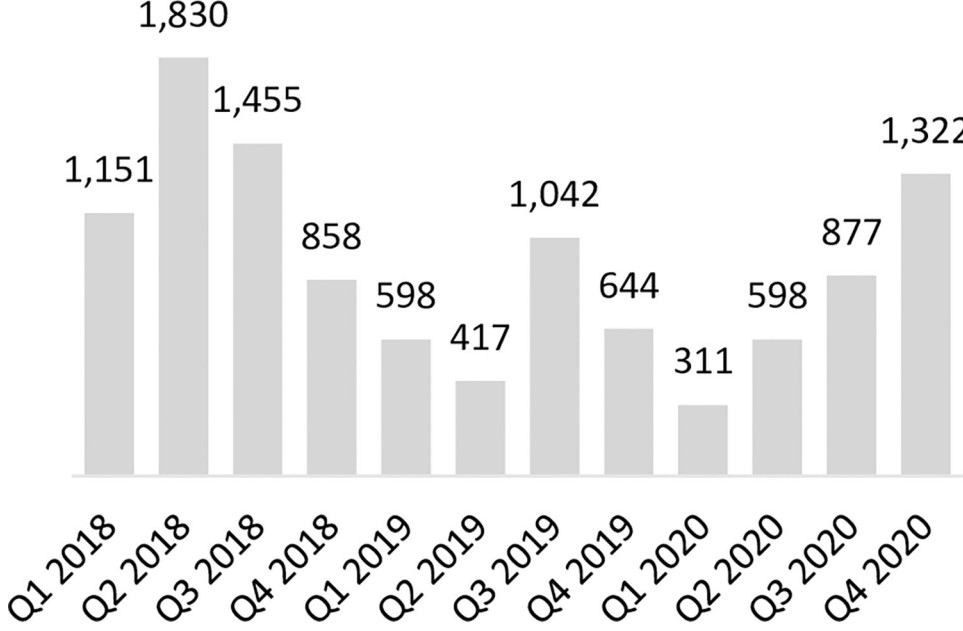

**Fig 2. Number of IUDs provided in program-supported sites in Uganda.**

had provided at least one IUD in the last 90 days. At endline, LARC services were routinely provided at more eligible facilities; 96% of focal facilities had provided at least one implant and 92% had provided at least one IUD in the last 90 days. Across the focal facilities providing LARCs, the number of implants provided increased significantly from 4,560 at baseline to 23,463 at endline, but the number of IUDs provided decreased slightly from 2,981 at baseline to 2,199 at endline (not statistically significant) (**Figs 2 and 3**). As IUD volumes remained

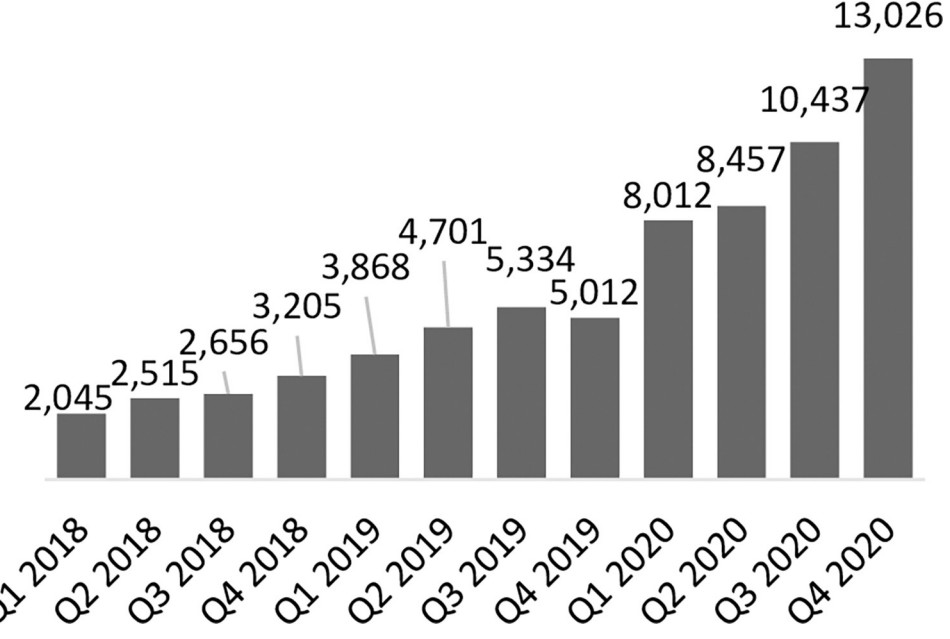

**Fig 3. Number of implants provided in program-supported sites in Uganda.**

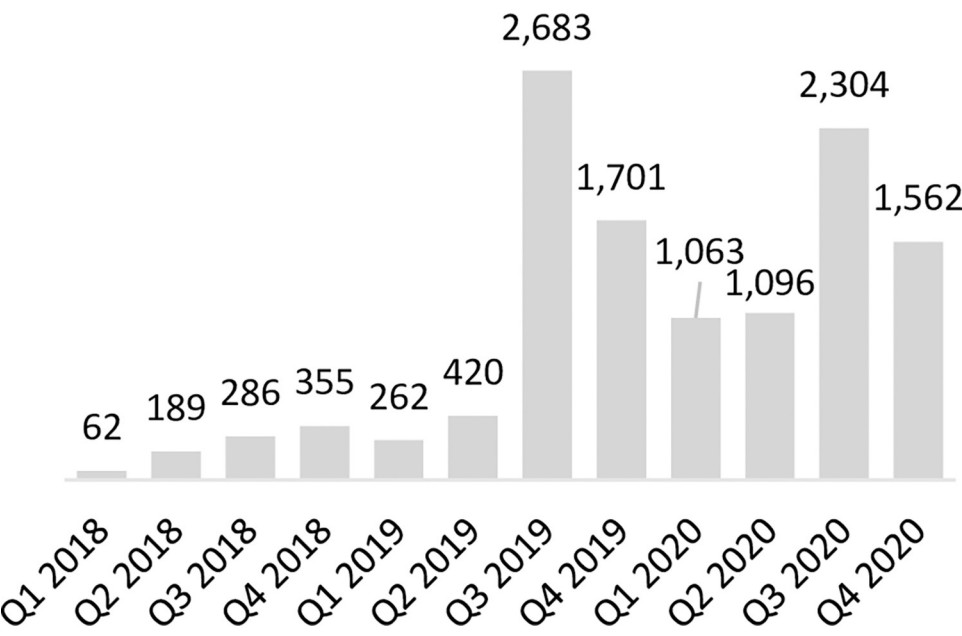

**Fig 4. Number of IUDs provided in program-supported sites in Zambia.**

largely stable, implant volumes increased significantly, thus skewing the share of CYPs generated by LARCs firmly towards implants. Specifically, the CYPs generated by implants contributed an increasing share to the CYP method mix, from 48% in 2018 to 82% by the end of 2020. Conversely the share of the CYP method mix contributed by IUDs decreased from 43% to 12% in the same period. The overall share of CYPs generated by LARCs stayed largely constant over time, increasing slightly from 91% to 94% from baseline to endline.

In Zambia, 54% of LARC-eligible focal facilities had provided at least one implant and 15% had provided at least one IUD in the last 90 days at baseline. At endline, 68% of LARC-eligible facilities had provided at least one implant and 32% had provided at least one IUD in the last 90 days. These increases in the number of facilities providing LARC services were reflected in total service volumes for both methods. Implant service volumes increased from 1,884 at baseline to 7,394 at endline while IUD service volumes increased from 251 to 3,866 in the same period (**Figs 4 and 5**). As service volumes increased, the number of CYPs generated by implants contributed an increasing share to the CYP method mix from 38% to 53% by the end of 2020, while IUD CYPs accounted for 6% of the method mix at baseline and 33% of the method mix at endline.

## Discussion

In both countries, the baseline assessment found notable gaps in the availability of the supplies and equipment needed to provide implant and IUD services, per current national clinical guidelines and protocols which provide guidance on the provision and quality assurance of LARC services. This was in contrast with the relatively high coverage of routine LARC provision at baseline. In both countries, INGO implementing partner mobile clinical outreach channels contributed to baseline service provision levels to an unknown degree. INGO outreach teams mobilized clients for services and travelled with trained clinicians and LARC supplies to provide LARC services per schedules agreed with facilities and local government. In Uganda, Marie Stopes International Uganda and Population Services International

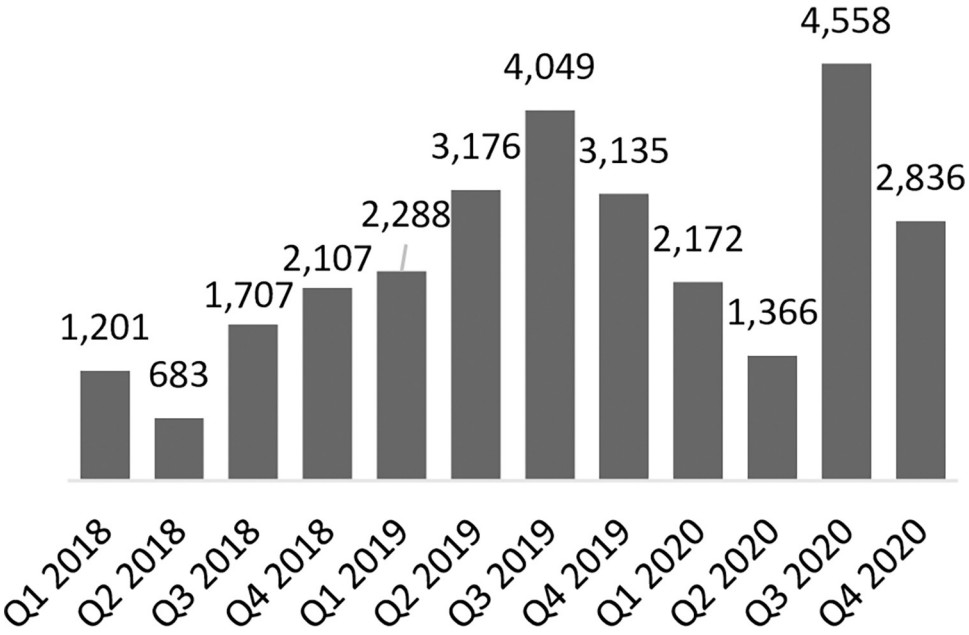

**Fig 5. Number of implants provided in program-supported sites in Zambia.**

implemented family planning outreach activities to serve hard-to-reach rural communities with LARC and other FP services prior to the start of the program, and intermittently throughout the program lifetime. In Zambia, Marie Stopes was active in program geographies, and at the time of baseline data collection had recently started reporting into public sector facility registers. Outreach services are not disaggregated in HMIS forms or DHIS2, so it is not possible to confirm the extent to which outreach contributed to service volumes at baseline or throughout the program.

It was anticipated however that by improving routine availability of supplies and equipment and by strengthening human resource capacity, public sector facilities would be better able to meet demand for LARC services, decreasing reliance on implementing partner outreach. In enabling public sector facilities to better meet demand, this would then grow demand further as more clients have positive experiences accessing LARC services from public sector providers. As demand for LARC products grows, anecdotal evidence suggests that health worker confidence to provide services would increase, reinforcing a virtuous cycle of client demand and service availability.

While the interventions implemented in both Uganda and Zambia were similar, the details of the context and results achieved were different, contributing to complementary learnings.

## Uganda

In Uganda, baseline facility assessments revealed significant gaps in availability of health workers trained to provide IUD services and supplies needed to provide IUD services. Assessments identified stock outs of ringed forceps, speculums, sponge holding forceps, and only 61% of facilities had actual IUDs in stock on the day of the assessment. Routine stock outs were attributed to inaccurate quantification, late product ordering and lack of earmarked budget at facilities or central government level to procure or replace missing equipment. Issues were noted around periodically low stock levels of IUDs due to delays in delivery of commodities to health centers and the fact that IUDs are currently excluded from the standardized basket of

commodities that health center III facilities receive quarterly. Routine disruption in availability of key supplies required for IUD services continued throughout the program lifetime and likely enhanced existing preferences for implant insertion amongst trained health workers in Uganda. At the start of the program, less than half (45%) of focal facilities had an IUD-trained health worker, and only about a third (31%) of facilities had benefited from on-the-job mentoring on IUD provision. In contrast, most facilities (84%) had an implant-trained provider, and over half (53%) had benefitted from previous mentorship. It was anecdotally reported that health workers were more confident in implant insertion manual skills as a result of having had more practical experience in insertion of implants than IUDs and relatively few opportunities to practice IUD manual insertion skills under supervision or mentorship. Consequently, they counselled clients on methods they were most comfortable providing and on methods they felt able to provide given supply-related stock outs.

The program in Uganda was successful in strengthening key systems necessary for routine IUD provision but was unable to establish a virtuous cycle for IUD uptake. The program's capacity building initiatives in collaboration with district health teams (described in Supporting Information File **Table 1**) led to nearly all (98%) focal facilities having a trained IUD provider, and due to improvements in accuracy of quantification, and more proactive emergency orders and redistribution of stock between focal facilities, most (65%) focal facilities had all the necessary supplies and commodities for IUD provision at endline. However, while the percent of program facilities that provided at least one IUD in the last 90 days increased from 73% to 92% by end of program, suggesting that IUD services were more consistently offered because of systems strengthening, the number of IUDs provided by focal facilities remained largely stable as seen in **Fig 2**. There was a non-statistically significant decrease of 26%, from 2,981 at baseline to 2,199 at endline, and within that time, IUD volumes dipped considerably lower at midline in 2019. Anecdotal feedback from government district health teams and CHAI program staff based in districts suggest that strong, prevalent myths and misconceptions circulating about IUDs, including linking IUD use to infertility and various cancers, as well as discomfort around how intra-uterine methods are inserted, meant that women were more inclined to choose implants as their preferred LARC option as both options became more routinely available. This likely contributed to a virtuous cycle for implant provision and uptake, where a greater number of satisfied users of public sector implant services emphasized the benefits of implants within their peer and social networks. Facilities were increasingly able to respond to existing demand for implants as the preferred LARC option, as demonstrated by a strong and largely consistent upward trend in implant services provided (**Fig 3**). As availability of high-quality implant services increased, more clients shared their positive experiences in their community, organically stimulating demand, which in turn strengthened health worker skills and confidence through frequency and repetition of implant insertion. This likely laid the groundwork for mobile clinical outreach teams deployed by Marie Stopes Uganda in program geographies to support lower-level facilities in meeting increased demand for implant services during periodic outreach events. Organic peer-to-peer dialogue and in-depth discussion between prospective clients, community members, and satisfied users was likely a critical enabler for implant mobilization and service volume growth. Conversely, IUD service provision was not stimulated in the same way. Prior to this public sector capacity building program, IUDs were largely provided during intermittent outreach events. Given the side effects non-hormonal IUDs are associated with, including menorrhagia, IUD users benefit from follow-up care and opportunities to discuss side effect management with health providers. However public sector health facilities were ill-equipped to provide routine follow-up counselling or to support side-effect management, and we hypothesize that limited capacity or readiness of public facilities to provide routine clinical follow up may have left contributed to negative user

experiences, which in turn contributed to negative narratives. In Uganda, more concerted effort, targeted community sensitization and demand generation interventions would have likely been needed to jump start demand and interrupt existing negative narratives circulating in communities surrounding IUDs for which there was not a critical mass of existing demand that could be met by improving service readiness and provision. Unfortunately, due to the COVID-19 pandemic and related government-issued guidance on social distancing and limiting large gatherings, the program was unable to support implementation of some planned community mobilization and sensitization activities. While program-supported radio spots and shows continued to promote the benefits of FP in general and LARCs throughout the pandemic, a coordinated campaign of more facilitated face-to-face sensitization events which leverage peer to peer mobilizers and satisfied users may have effectively tackled entrenched myths and misconceptions about IUDs in target geographies, but opportunities for this were limited.

## Zambia

In Zambia, baseline assessments confirmed significant gaps in contraceptive commodities. Availability of implants and IUDs continued to vary significantly throughout the program in Zambia due to insufficient government funding for commodity procurement, exacerbated by the mounting fiscal pressures of the COVID-19 pandemic response. Severe fiscal challenges across government ministries led to reductions in national health programs and insufficient funding for the national drug budget. Routine stockouts of LARC commodities as well as other LARC-related supplies were also attributed to lack of sufficient funding for transport between central level warehouses, district hubs and facilities, making it difficult to deliver products to districts and facilities per set schedules. Exacerbating these challenges, Zambia (as well as Uganda) was impacted by global level manufacturing capacity constraints for contraceptive implants, which pre-dated and continued throughout the program, further contributing to inconsistent availability and consumption of one-rod implant Implanon NXT and two-rod implant Jadelle, as well as contributing to challenges in accurate facility-level forecasting for implants. Finally, inconsistent documentation of services provided by health workers within facility service registers in Zambia made it difficult for facilities and districts to interpret past consumption and create data-based forecasts to inform future orders. These significant financing and supply chain systems challenges resulted in variability in availability of implant and IUD commodities; whereas 86% of eligible facilities had implants in stock on the day of the assessment at baseline, only 62% of focal facilities had implants in stock at endline, and there was no notable change in IUD stock availability over the course of the program. Commodity stock outs therefore likely had a dampening effect on overall service volume growth observed. Despite this, the program in Zambia was able to improve routine availability and provision of both implant and IUD services at focal facilities as observed through the increase in facilities providing implants (from 46 to 51 facilities) and IUDs (from 13 to 33 facilities) within the last 90 days from baseline to midline. This was accomplished by addressing systems-level barriers to service provision including capacity of health workers. At baseline only 60% and 54% of focal facilities had providers trained on implant and IUD provision respectively; this increased to 92% at endline. Trained providers then received government-led on-the-job mentoring to support integration of skills into service delivery.

As in Uganda, the program in Zambia supported enhanced readiness of facilities to provide services. Awareness of LARC options and benefits amongst potential users improved, which launched a virtuous cycle of supply and demand for LARC methods, demonstrated by a steady upward trend in implant and IUD services provided by focal facilities over time as seen in **Figs**

4 **and** 5. A notably significant increase in both IUD and implant volumes was observed in Q3 2019, when Zambia received an influx of LARC commodities from global procurers, and service provision volumes continued to increase over time, though volumes were variable due to the impacts of commodity stock outs. We believe this demonstrates that health worker capacity building and systems strengthening efforts were largely successful in increasing readiness to provide the service amongst focal facilities, enabling facilities to respond to existing and growing demand for these services (subject to commodity availability).

In contrast to Uganda, the program in Zambia appeared to have kick-started a virtuous cycle of supply and demand for IUDs. There are important points of difference to note between the Uganda and Zambia contexts which may help explain this. First, baseline IUD service provision and associated volumes reported were relatively low in Zambia providing more scope for growth; per program data, only 15% (13/85) of eligible facilities routinely reported providing IUDs at the start of the program. Second, the percent of women who have ever heard of the IUD as a contraceptive method was relatively low in program geographies in Zambia at baseline; according to Zambia's 2018 Demographic and Health Survey, in Northern Province, only 51.2% of women were aware of the IUD as a contraceptive, which is low compared to awareness at the national level in Zambia (63.6%) [19] and as compared to awareness of IUDs in the program regions of North Buganda and Bunyoro in Uganda (87.2%) [20]. Anecdotal reports from stakeholders indicated that in Uganda, previous IUD service provision by outreach teams without investment in public sector capacity to provide IUD side effect management and follow up care may have contributed to a cycle of low IUD provision and use. By contrast, in Zambia, health workers and users had limited experiences, both positive and negative, with IUDs. Strong opinions against the method amongst community members and key opinion leaders may have therefore been less fully formed. Over the program lifetime the percent of facilities that had provided IUDs within the last 90 days doubled to 32%, and the average monthly consumption of IUDs increased from 42 to 644 by endline. From July 2019, following an influx of LARC commodities to Zambia, INGO Marie Stopes Zambia restarted implementation of partner outreach events after having paused due to commodity shortages and in similarity with Uganda, mobile clinical outreach units deployed by INGOs to program geographies also responded to and met increased demand for LARC services generated by government health workers and community mobilization structures.

## Role of COVID-19 pandemic

With regards to the impact of the COVID-19 pandemic on program implementation and results, as observed in Table 2, by the end of 2019, immediately prior to the start of the COVID-19 pandemic, both programs had observed significant increases in the proportion of LARC-eligible focal facilities with a provider trained to provide implants and IUDs and in the proportion of focal facilities with tracer commodities and equipment for implant and IUD services, with the caveat that in Zambia, IUD and implant provision dipped from Q4 2019 primarily due to fiscal challenges which led to stockouts of LARC commodities. Following improvements in service readiness, from midline focal facilities reported significant growth in service volumes, with the exception of IUD services in Uganda. It should be noted that from March 2020, the global COVID-19 pandemic increasingly impacted health service provision and health seeking behavior in both countries. To mitigate the impact of the pandemic and sustain previous gains in service readiness, both programs worked with subnational government teams to activate informal channels for communication, enabling remote mentorship through instant messaging applications during which government mentors reinforced the importance of continuity of LARC service provision and facility managers solved for LARC-

related supply issues in real time. Apart from increased stock outs of implants and IUDs in Zambia in 2020, both programs maintained and built on progress made pre-pandemic to increase readiness of focal facilities to provide LARC services. In both countries, to mitigate the impact of the pandemic on demand for routine contraceptive services, programs supported the procurement of Personal Protective Equipment (PPE) for community health workers to enable them to promote hand washing, social distancing, and mask wearing while continuing to mobilize demand for contraception. Community health workers were provided with information to allay community fears related to visiting health facilities and perceived increased risk of catching COVID-19.

## Recommendations

Based on the experience of this program, we believe that systems-strengthening to improve service readiness is a key input to establishing a virtuous cycle for LARCs where client demand and provider skills, comfort, and confidence are mutually reinforcing. Capacity building to improve service readiness must be rationally sequenced with interventions to ensure supply and increase client demand for LARC methods so as to support health worker skills development and integration into routine service delivery. When a virtuous supply and demand cycle is in place, there is: accurate information on LARCs circulating amongst communities, increasing client demand; high provider skills acquisition, skills retention and buy-in; high provider comfort and confidence to counsel or provide the service; accurate, high-quality counseling by health workers; and consistent supply available of LARC commodities. Information and experiences are shared by word of mouth from one satisfied or dissatisfied user to her peers, and it is critical that LARC scale-up interventions are designed to interrupt negative narratives and establish positive narratives surrounding both implants and IUDs. This is particularly true for IUDs, for which intransigent myths and misconceptions related to risks of infertility, cancer, and other long-term health consequences significantly impact demand, as observed in Uganda.

There is increasing interest in and recognition of the role of the public sector in increasing access to and equity in family planning services [21]. Implementation experience suggests that in order to rapidly improve the readiness of public sector facilities to provide LARC services, a number of initial investments are likely to be needed to catalyze the cycle of supply and demand for implants and IUDs. Both programs used donor funding to enable government-led training and mentorship and procure and distribute missing equipment needed for LARC service provision which resulted in rapid activation of facilities as access points for LARC services. Catalytic interventions, including traditional implementing partner supported mobile clinical outreach, can and should be linked to broader systems strengthening interventions, and where possible should seek to complement or help to facilitate commitment of subnational government resources to the implementation of program interventions. In both Uganda and Zambia, subnational government leadership have now integrated LARC clinical mentorship into SRMNH operational plans and budgets, with local governments making in-kind and financial commitments to the continuity of clinical mentorship. This suggests that subnational government structures and public health facilities may be able to take on more of the contraceptive-related program activities traditionally led by implementing partners and funded by international donors, integrating these to some extent into domestic budgets and plans. Strengthening government-owned supply chain systems to ensure availability of LARC commodities at facilities, while in tandem strengthening training and mentoring for health workers, is essential for enabling increased LARC service provision and use in the public sector. In parallel, barriers to demand must be meaningfully addressed, and further operational research to demonstrate

cost-effective, sustainable, and scalable approaches to community mobilization and social and behavior change communication is needed to inform government and partner approaches as health system capacity increases. We recommend drawing from program learnings around the importance of using a systems lens to create a virtuous cycle of supply and demand when implementing aimed at increasing availability and voluntary uptake of LARCs and other family planning services in Uganda, Zambia, and comparable market contexts.

## Limitations

There are several limitations to this analysis and discussion of the program. The service delivery data reported by facilities includes LARC services provided at each site by public sector staff as well as by visiting INGO teams. It is not possible to ascertain the proportion of LARC services directly provided by government health workers versus INGO mobile clinical outreach teams, however INGO activities and investments were not reported to change substantially over this program period. Due to the COVID-19 pandemic, facility assessments were largely conducted over the phone in 2020 and facilities self-reported data. However, as there were no direct benefits or risks of honest reporting, we believe that this presents minimal risk of bias. The definition for the indicator that measured the percent of facilities with tracer commodities and equipment for IUDs excluded uterine sounds from facility assessment checklists. Uterine sounds are used where transvaginal ultrasound is not available to ascertain uterine depth and this equipment is required for routine IUD insertion. Availability of uterine sounds for IUD provision in focal facilities is therefore unknown. In Uganda, data for the indicator that measured the percent of focal facilities with a trained and mentored provider in IUD provision was not collected at baseline; baseline data used represents the proportion of focal facilities with a provider trained and mentored in post-partum IUD provision. With respect to service provision data, the program reviewed and cleaned data pulled from HMIS, investigating any outliers. However, HMIS data may be subject to unknown reporting errors. In routine HMIS data, it is not possible to separate the services provided through INGO outreach from routine service provision at facilities. Facility catchment areas and related number of households served changed during the program due to changes in local administrative zones in Uganda and Zambia, although the positive trend in overall service growth suggests the program largely mitigated the impact of these changes. Lastly, both implants and IUDs can be inserted immediately post-partum. Both programs aimed to increase access to post-partum LARC services through improving quality of health education and counselling during antenatal care and documentation within client records to indicate choice of optional post-partum contraception, however post-partum LARC results and learnings are outside the scope of this paper.

## Conclusions

Technical assistance partners, local government units, and health managers must work together to improve the availability of LARC products and services in eligible public sector facilities. Rapid increases in readiness of facilities to provide services can be achievable when a combination of catalytic and systems strengthening interventions are deployed. Whereas traditional investments in LARC scale-up have leveraged INGO service delivery partners to fill gaps in public sector capacity and personnel to provide LARCs, this program in Uganda and Zambia demonstrated that the public sector can be rapidly capacitated to provide LARC services. Implemented collaboratively, partner-supported outreach channels may be able to help establish a virtuous cycle of LARC provision in the public sector by supporting the public sector to meet rapidly growing demand for LARC services. With respect to IUD scale up however, it is

important that catalytic INGO outreach is implemented in tandem with public sector health systems strengthening, to ensure public sector facilities can provide high quality counselling and management as part of follow up care. By supporting clients with management of side effects, public sector facilities can support positive client IUD experiences, and contribute to positive narratives about IUDs in communities. To increase public sector capacity for LARC service provision, government systems must be strengthened to bolster health workforce management, health worker capacity building, supply chain management, and community mobilization simultaneously.

## Supporting information

**S1 Table. Key LARC-related program activities.** This table includes the LARC-related activities implemented by government and CHAI in Uganda and Zambia in this program.
(DOCX)

**S1 File. Uganda datasets.**
(XLSX)

**S2 File. Zambia datasets.**
(XLSX)

## Acknowledgments

We would like to express our deep gratitude to the women, families, and health workers of Kagadi, Kakumiro, Kassanda, Kibaale, Mityana, and Mubende districts in Uganda and Northern Province in Zambia for their engagement in this work. This paper reflects effort from countless individuals, but we would like to recognize the partnership and contributions of key Ministry of Health staff that supported implementation of this program: Dr. Jesca Nsungwa Sabiiti in Uganda and Beauty Muntanga, the late Lissah Susiku, Dr. Lawrence Phiri and Maxwell Kasonde in Zambia. We would also like to thank CHAI staff that were invaluable in developing strategy and implementation of the program: Rosette Birungi, Manish Burman, Ronald Kizito, Refilwe Kotane, Kelly McCrystal, Robyn Churchill, Waza Mhango, Levy Mkandawire, Helen Mwiinga, Racheal Najjemba, Flavia Namayengo, Mindy Scibilia, Monica Setaruddin, Margaret Siame, Andrew Storey, Lawrence Were, and Rabson Zimba.

## Author Contributions

**Conceptualization:** Aniset Kamanga, Micheal Lyazi, Margaret L. Prust.

**Data curation:** Aniset Kamanga, Micheal Lyazi, Margaret L. Prust, Naomi Medina-Jaudes, Lupenshyo Ngosa, Margaret Nalwabwe.

**Formal analysis:** Aniset Kamanga, Micheal Lyazi, Margaret L. Prust, Naomi Medina-Jaudes, Lupenshyo Ngosa, Margaret Nalwabwe, Emma Aldrich.

**Funding acquisition:** Caitlin Glover.

**Investigation:** Aniset Kamanga, Micheal Lyazi, Margaret L. Prust, Naomi Medina-Jaudes, Lupenshyo Ngosa, Margaret Nalwabwe, Martha Ndhlovu.

**Methodology:** Aniset Kamanga, Micheal Lyazi, Margaret L. Prust.

**Project administration:** Dynes Kaluba, Angel Mwiche, Richard Mugahi, Joy Batusa, Morrison Zulu, Andrew Musoke, Hilda Shakwele.

**Resources:** Aniset Kamanga, Micheal Lyazi, Margaret L. Prust, Naomi Medina-Jaudes, Lupenshyo Ngosa, Margaret Nalwabwe, Martha Ndhlovu.

**Software:** Aniset Kamanga, Micheal Lyazi, Margaret L. Prust, Naomi Medina-Jaudes, Lupenshyo Ngosa, Margaret Nalwabwe.

**Supervision:** Joy Batusa, Morrison Zulu, Andrew Musoke, Hilda Shakwele.

**Validation:** Aniset Kamanga, Micheal Lyazi, Margaret L. Prust, Naomi Medina-Jaudes, Lupenshyo Ngosa, Margaret Nalwabwe.

**Visualization:** Margaret L. Prust.

**Writing – original draft:** Aniset Kamanga, Micheal Lyazi, Margaret L. Prust, Naomi Medina-Jaudes, Lupenshyo Ngosa, Emma Aldrich.

**Writing – review & editing:** Aniset Kamanga, Micheal Lyazi, Margaret L. Prust, Naomi Medina-Jaudes, Lupenshyo Ngosa, Margaret Nalwabwe, Martha Ndhlovu, Dynes Kaluba, Angel Mwiche, Richard Mugahi, Joy Batusa, Morrison Zulu, Andrew Musoke, Hilda Shakwele, Caitlin Glover, Emma Aldrich.

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
