## [Decision Letter · Decision Letter 0]

29 Mar 2023

PONE-D-23-00410Strengthening systems to provide Long-Acting Reversible Contraceptives (LARCs) in public sector health facilities in Uganda and Zambia: Program results and learningsPLOS ONE

Dear Dr. Aldrich,

Thank you for submitting your manuscript to PLOS ONE. After careful consideration, we feel that it has merit but does not fully meet PLOS ONE’s publication criteria as it currently stands. Therefore, we invite you to submit a revised version of the manuscript that addresses the points raised during the review process. Please submit your revised manuscript by May 13 2023 11:59PM. If you will need more time than this to complete your revisions, please reply to this message or contact the journal office at plosone@plos.org. Please include the following items when submitting your revised manuscript:A rebuttal letter that responds to each point raised by the academic editor and reviewer(s). You should upload this letter as a separate file labeled 'Response to Reviewers'.A marked-up copy of your manuscript that highlights changes made to the original version. You should upload this as a separate file labeled 'Revised Manuscript with Track Changes'.An unmarked version of your revised paper without tracked changes. You should upload this as a separate file labeled 'Manuscript'.

We look forward to receiving your revised manuscript.

Kind regards,

Lea Sacca

Academic Editor

PLOS ONE

Journal Requirements:

2. Please include a complete copy of PLOS’ questionnaire on inclusivity in global research in your revised manuscript. Our policy for research in this area aims to improve transparency in the reporting of research performed outside of researchers’ own country or community. The policy applies to researchers who have travelled to a different country to conduct research, research with Indigenous populations or their lands, and research on cultural artefacts. The questionnaire can also be requested at the journal’s discretion for any other submissions, even if these conditions are not met.  

Please find more information on the policy and a link to download a blank copy of the questionnaire here:

https://journals.plos.org/plosone/s/best-practices-in-research-reporting.

Please upload a completed version of your questionnaire as Supporting Information when you resubmit your manuscript.

4. Thank you for stating in your financial disclosure:  

"This work was made possible with financial support from The ELMA Foundation and an anonymous donor (grants received and managed by HS and AM). The views expressed in this report are the opinions of the authors, and do not necessarily reflect the official policies of the funders. The funders had no role in study design, data collection and analysis, decision to publish, or preparation of the manuscript."

PLOS ONE requires you to include in your manuscript further information about the funder so that any relevant competing interests can be assessed. Please respond to the following questions:

1) Please state whether any of the research costs or authors' salaries were funded, in whole or in part, by a tobacco company (our policy on tobacco funding is at http://journals.plos.org/plosone/s/disclosure-of-funding-sources)  

2) Please state whether the donor has any competing interests in relation to this work (see http://journals.plos.org/plosone/s/competing-interests) . 

3) Please state whether the identity of the donor might be considered relevant to editors or reviewers’ assessment of the validity of the work.

4) If the donors have no perceived or actual competing interests, please state: “The authors are not aware of any competing interests”. 

This information should be included in your cover letter. We will amend your financial disclosure and competing interests on your behalf.

5. Please ensure that you refer to Figures 2 to 5 in your text as, if accepted, production will need this reference to link the reader to the figure.

Reviewers' comments:

Reviewer's Responses to Questions

**Comments to the Author**

1. Is the manuscript technically sound, and do the data support the conclusions?

Reviewer #1: Yes

Reviewer #2: Yes

2. Has the statistical analysis been performed appropriately and rigorously? 

Reviewer #1: Yes

Reviewer #2: Yes

3. Have the authors made all data underlying the findings in their manuscript fully available?

Reviewer #1: Yes

Reviewer #2: Yes

4. Is the manuscript presented in an intelligible fashion and written in standard English?

Reviewer #1: Yes

Reviewer #2: Yes

5. Review Comments to the Author

Reviewer #1: Comments to the Author

Abstract

• Give a brief introduction/definition and examples of the Long acting reversible contraceptives (LARCs) in Uganda and Zambia

• Define readiness since it’s a key word under methods

• Under methods, quote which indicators were monitored and why? Is that they are recommended by MoH/WHO/CDC?

Introduction

• Begin with defining unintended pregnancies at the beginning

• Show the commitment of Uganda and Zambia to increase access to family planning services either by MOH policies or strategic plans. Quote and cite them

• Give a paragraph describing and discussing supply-demand factors before paragraph two

• Before describing the efforts of NGOs (paragraph 3), talk about government efforts to promote access and utilization of LARCs. This should preceed before the INGOs efforts

• Give a full paragraph describing this programme which was implemented in Uganda and Zambia. Be clear on how it supplemented Public/government of Uganda efforts, quote the indicators which we were interested in but also the interventions which were undertaken

Methods

• Describe the districts/hospitals that participated

• Mention the number(s) of interventions and clearly show a distinction beyond government efforts because you stand a risk of failing to attribute the results we see to your efforts

• Describe how different this LARC programme was designed, implemented, monitored. Make these sections clearly stated

• Under setting, give the population profiles for Uganda and Zambia. Quote the implementing efforts/initiatives in the two countries like Maries Stopes, UNFPA and so forth. Quote the statistics as much as you can. You can cite the UDHS survey 2022

• For baseline assessment, clearly mention how it was done, by who? Which indicators did we focus on? How did we measure this? Which tools did we use? What informed this baseline? Did we use standard tools? Etc

• Mention the names of IRBs that reviewed your protocols and dates of approvals

Results

• Try to have some factors that could explain the high uptake in Zambia than in Uganda? Try to run factors associated and we could see what to leverage on from Zambia for Uganda

• For us to make a good case, try to show the statistics of our indicators three years before our intervention such that we can see a change during your intervention period

Discussion

• Break the first paragraph into different sections. It’s long. Begin with a preamble of what we found out

• Support your assertions with existing international literature like USA, UK,China etc

• Discuss the contributions of other NGOs like Marie Stopes, UNFPA, USAID RHITES and so forth

• Align with MoH Policies and guidelines

Conclusions and recommendations

• Clearly give these findings with MoH guidelines/strategic plans/policies

• What are you telling implementing partners on your results

• What does this imply for National Medical stores and Joint Medical stores

• What are the areas for future research?

• What are the implications of these results to SDGs and national strategic plans in the two countries

Reviewer #2: Strengthening systems to provide Long-Acting Reversible Contraceptives (LARCs) in public sector health facilities in Uganda and Zambia: Program results and learnings.

Summary.

The authors conducted a study looking at strengthening systems to provide Long-Acting Reversible Contraceptives (LARCs) in public sector health facilities in Uganda and Zambia: Program results and learnings.

This study is significant and applicable to both study settings as there is a high rate of unintended pregnancies. The article is generally well written, but the following points need to be considered:

1. The contextual differences or similarities between the two countries/study sites should be explained. This is crucial because it will enable a better understanding of the variations in project intervention uptake, such as the use of intrauterine devices (IUDs). The discussion's elaborated contextual explanations are insufficient. The authors might also consider other social and cultural aspects that have been covered in other works of literature.

2. The authors ought to explain why particular provinces in the two countries were chosen. How does the provision of sexual, reproductive, maternal, and new-born health (SRMNH) in each nation compare on a provincial level?

3. The authors must describe the type/category of health professionals trained at these facilities. Discuss how the project dealt with staff attrition throughout the course of the study as well.

Minor comments

1. Provide a reference for Figure 1. Virtuous cycle of LARC availability and provision.

2. In the discussion section first paragraph “In Zambia, Marie Stopes Zambia was active in Zambian program geographies, and at the time of baseline data collection had recently started reporting into public sector facility registers.” Rephrase to avoid text redundancy

6. PLOS authors have the option to publish the peer review history of their article (what does this mean?). If published, this will include your full peer review and any attached files.

Reviewer #1: **Yes: **Abel Wilson Walekhwa

Reviewer #2: No

---

## [Author Response · Author response to Decision Letter 0]

27 Jun 2023

Reviewer #1 comments 

Abstract

1. Give a brief introduction/definition and examples of the Long acting reversible contraceptives (LARCs) in Uganda and Zambia.

Response: Thank you for this suggestion. We added the following underlined text to the first sentence of the abstract to further define LARCs. 

“In Uganda and Zambia, both supply- and demand-side factors hamper availability of long-acting reversible contraceptives (LARCs), including implants and intrauterine devices (IUDs), at public sector facilities.”

2. Define readiness since it’s a key word under methods

Response: Thank you for this point. We’ve defined readiness further in the Key Indicators section with the following text: 

“We define readiness as the presence of the necessary commodities, equipment, and trained staff to provide a specific service”

3. Under methods, quote which indicators were monitored and why? Is that they are recommended by MoH/WHO/CDC?

Response: In an effort to maintain the journal word limit for the abstract, we have not described the indicators in detail in the abstract; however, the indicators are listed in Table 1. We also added the following underlined text to respond to the request for clarification about who agreed to or recommended the indicators:

“In relation to LARCs, several key indicators were agreed upon at the outset of the program by program and government staff as a foundation to align all stakeholders around strategic goals and measures of success (Table 1).”

Introduction

4. Begin with defining unintended pregnancies at the beginning

Response: We have added the following sentence to define unintended pregnancies: 

“Unintended pregnancies are pregnancies that are mistimed, unplanned or unwanted at the time on conception.” 

5. Show the commitment of Uganda and Zambia to increase access to family planning services either by MOH policies or strategic plans. Quote and cite them

Response: Thank you for this comment. We have added the following underlined text to demonstrate the commitment of both governments to expanding access to family planning services:

“In Uganda and Zambia, high numbers of unintended pregnancies contribute to high rates of morbidity and mortality due to unsafe abortion and lack of access to routine and emergency obstetric care. The governments of Uganda and Zambia are committed to increasing access to and use of modern family planning (FP) to reduce unintended pregnancies and maternal morbidity and mortality. For example, the Zambian National Health Strategic Plan of 2022 – 2026 commits to increase the proportion of women of reproductive age (aged 15–49 years) who have their need for family planning satisfied with modern methods from 68.5% in 2018 to 80% 2026 [1], and in Uganda, government seeks to increase the modern contraceptive (mCPR), for all women, from 30.4% in 2020 to 39.6% by 2025 and reduce the unmet need for modern contraception from 17% in 2020 to 15% by 2025[2]”

6. Give a paragraph describing and discussing supply-demand factors before paragraph two

Response: Thank you for highlighting this. The supply and demand side issues are described as follows:

“Public sector supply-side gaps include insufficient numbers of trained health workers skilled to provide LARC services, persistent gaps in LARC commodities at service delivery points, and lack of essential equipment, medicines and supplies to provide LARCs. At the same time, demand-side issues related to family planning include lack of awareness of LARC benefits and pervasive myths and misconceptions about implants and IUDs limit demand for family planning…”

7. Before describing the efforts of NGOs (paragraph 3), talk about government efforts to promote access and utilization of LARCs. This should precede before the INGOs efforts

Response: We have added information about the key government commitments in the family planning space that provide a foundation for all programming. The additional text is as follows:

“The Zambian and Uganda governments are working to address these issues and expand access to family planning, including LARCs. In Zambia, the government pledged to invest $12 million in family planning programs in 2023, with yearly increases of 30% after that [13]. In Uganda, the government has committed to ring fence 50% of the domestic reproductive health commodities budget for procurement, storage and distribution of FP commodities from by 2025 [14].”

8. Give a full paragraph describing this programme which was implemented in Uganda and Zambia. Be clear on how it supplemented Public/government of Uganda efforts, quote the indicators which we were interested in but also the interventions which were undertaken

Response: Thank you for highlighting this. The purpose of Supplemental Table 1 is to provide detailed descriptions of the program activities. Upon reflection on this comment, we realized that the reference to S1 Table was perhaps not descriptive or clear enough to guide readers, so we have added the following text to provide more information. This text also addresses the question of how these activities supplement the governmental efforts. 

“The program activities relating to LARCs that were implemented in Uganda and Zambia by CHAI and the government are described in detail in S1 Table. Though the activities were implemented with government collaboration, these activities were beyond and in addition to the routine service support activities performed by the government.”

Methods

9. Describe the districts/hospitals that participated

Response: Thank you for this comment. We agree that more contextual information is useful, and we have added the following text in the Setting section in response:

“These areas of Zambia and Uganda were selected by the government as the focal area for this work because they were some of the most underperforming areas of the country on SRMNH-related outcomes, Northern Province, Zambia is a very rural and hard-to-reach areas with no other maternal and newborn health programs active in the province at the time that this program was launched. The selected districts in Uganda area are within several hours drive from the capital of Uganda, Kampala, but nonetheless include areas that are hard-to-reach due to poor road networks.”

10. Mention the number(s) of interventions and clearly show a distinction beyond government efforts because you stand a risk of failing to attribute the results we see to your efforts

Response: Thank you for this comment, which seems closely related to the point above in comment #8. As noted above, we have added the following text to guide readers to S1 Table where the program activities are described in more detail. 

“The program activities relating to LARCs that were implemented in Uganda and Zambia by CHAI and the government are described in detail in S1 Table. Though the activities were implemented with government collaboration, these activities were beyond and in addition to the routine service support activities performed by the government.”

11. Describe how different this LARC programme was designed, implemented, monitored. Make these sections clearly stated

Response: We have added the following text to provide more information about the program:

“The program was designed by CHAI and the MOHs, building on previous experience of both groups and with careful consideration and adaptation for the local context and opportunities. CHAI developed a standardized mentorship framework and implementation package to assist focal geographies to address systems and facility-level barriers to SRH access, provision, and uptake. This assisted in addressing the gaps in LARC service provision. CHAI leveraged the pool of family planning government trainers and mentors to build health care worker capacity and drive acknowledgement amongst community traditional leadership of the need to scale up awareness of the benefits of family planning.”

12. Under setting, give the population profiles for Uganda and Zambia. Quote the implementing efforts/initiatives in the two countries like Maries Stopes, UNFPA and so forth. Quote the statistics as much as you can. You can cite the UDHS survey 2022

Response: Thank you for these questions. We agree that national statistics provide critical context for the program and results. National statistics on contraceptive use are provided in the introduction section. With regards to activities of other implementing partners, we describe the activities of Marie Stopes International in both Uganda and Zambia and of Population Services International in Uganda within the Discussion section. 

13. For baseline assessment, clearly mention how it was done, by who? Which indicators did we focus on? How did we measure this? Which tools did we use? What informed this baseline? Did we use standard tools? Etc

Response: Thank you for these questions. We have added the following underlined text to the Methods section to clarify the approach to the facility assessments:

“Facility assessments were conducted in Uganda and Zambia to determine which facilities had the resources and capacity to provide LARCs and other SRMNH services. The facility assessments were conducted by program staff or contractors using tools developed by the program to track the intended program indicators (Table 1) on family planning and other topics. The baseline assessment was conducted in July and August 2018 for Uganda and Zambia, respectively. ...”

14. Mention the names of IRBs that reviewed your protocols and dates of approvals

Response: This paper covers data from two different sources and the review process was tailored for each. The facility assessment process where primary data was collected by this program was reviewed by IRBs in both countries, as described in this text. We have added the date of review in response to this comment:

“The facility assessments were approved by ethical review committees in both countries in July 2018, including the ERES Converge Institutional Review Board (IRB) in Zambia (reference number 2018-Jul-017) the AIDS Support Organization IRB in Uganda (reference number 026/18-UG-REC-009). The IRBs determined that this work did not engage human subjects and that patient consent was not required.”

The routine service delivery data in aggregate data and its analysis was determined not to engage human subjects. We added the following text to clarify this:

“This data is available in aggregate form without any individualized patient information, and as a result, this work was determined by CHAI’s internal Scientific and Ethical Review Committee to not engage human subjects or require patient consent.”

Results

15. Try to have some factors that could explain the high uptake in Zambia than in Uganda? Try to run factors associated and we could see what to leverage on from Zambia for Uganda

Response: This point raises an important issue that we have discussed at length internally and would like to shed light on in the paper. We feel that this type of consideration of the results is more appropriate though for the Discussion section as opposed to the Results. The paragraph in the Discussion that begins with the following is an effort to consider the factors that made the experience of these two countries different.

“In contrast to Uganda, the program in Zambia appeared to have kick-started a virtuous cycle of supply and demand for IUDs. There are important points of difference to note between the Uganda and Zambia contexts which may help explain this.”

16. For us to make a good case, try to show the statistics of our indicators three years before our intervention such that we can see a change during your intervention period

Response: We agree that tracking trends over a longer period of time is preferable, but unfortunately, it is not possible in this case. The data from the facility assessments was collected by the program, beginning with the baseline assessment at the start of the program. We are not aware of any other dataset that tracks similar data on a consistent basis across the program-supported sites. The routine service delivery data is captured by facility staff in an on-going way, but during the program period, program staff were regularly engaging with government officials to review data and perform quality checks. We routinely identified data entry errors in the DHIS2 data system and program staff proactively followed MOH guidelines to have errors investigated and corrected. Because this data quality assurance process was performed only on data from 2018 forward, the data from prior years would not be comparable and may be misleading. 

Discussion

17. Break the first paragraph into different sections. It’s long. Begin with a preamble of what we found out

Response: We have broken out the first paragraph of the discussion into two smaller paragraphs and we hope that this improves readability. 

18. Support your assertions with existing international literature like USA, UK, China etc

Response: We appreciate and resonate with this comment. We reviewed available literature for comparable market contexts, specifically low-income, low-resource settings in Africa. Public sector health systems strengthening interventions with a focus on systems for delivering long-acting reversable contraception are under-documented in the relevant contexts. We reflected this in the Introduction where we note that neither implementation experiences nor evaluations of the effectiveness of public sector strengthening programs have been widely documented.

19. Discuss the contributions of other NGOs like Marie Stopes, UNFPA, USAID RHITES and so forth

Response: Thank you for this comment. We agree it is important to recognize the contributions of other implementing partners. UNFPA is active in both Uganda and Zambia at the national level, supporting among other things national supply chain program management. UNFPA has not be an active implementer at the subnational level in program geographies in either country. We describe the activities of Marie Stopes International and Population Services International in the Discussion section: ‘’In Uganda, Marie Stopes International Uganda and Population Services International implemented family planning outreach activities to serve hard-to-reach rural communities with LARC and other FP services prior to the start of the program, and intermittently throughout the program lifetime. In Zambia, Marie Stopes was active in program geographies, and at the time of baseline data collection had recently started reporting into public sector facility registers. Outreach services are not disaggregated in HMIS forms or DHIS2, so it is not possible to confirm the extent to which outreach contributed to service volumes at baseline or throughout the program.’’ The program focal districts in Uganda were not covered by USAID RHITES. 

20. Align with MoH Policies and guidelines

Response: Thank you for this important comment and point. Eligibility and requirements for the provision of LARC services by the public sector are outlined in the national Uganda Clinical Guidelines and national Zambia Family Planning Guidelines and Protocols documents, as described in the Setting section. Program and government staff aligned at the outset of the program on strategic goals and measures of success (Table) which was informed by the eligibility criteria for LARC provision as set out in national guidelines. The data presented and discussed in the paper is therefore a reflection on the health system’s capacity to meet standards and guidelines and provide quality LARC services in line with existing government guidance. We appreciate this may not have been clear to the reader, and have added the underlined text to the Discussion section: ‘’In both countries, the baseline assessment found notable gaps in the availability of the supplies and equipment needed to provide implant and IUD services, per current national clinical guidelines and protocols which provide guidance on the provision and quality assurance of LARC services.’’

Conclusions and recommendations

21. Clearly give these findings with MoH guidelines/strategic plans/policies? What are you telling implementing partners on your results? What does this imply for National Medical stores and Joint Medical stores? What are the areas for future research? What are the implications of these results to SDGs and national strategic plans in the two countries

Response: Thank you for these questions. Results and recommendations were widely disseminated through national technical working groups chaired by MOHs and attended by key implementing partners at the conclusion of the program. At the subnational level, provincial/district leadership reviewed program results and aligned on changes to local government operational plans and budgets. Key recommendations shared with implementing partners and recommendations taken forward by subnational governments are described in the Recommendations section: ‘’Implementation experience suggests that in order to rapidly improve the readiness of public sector facilities to provide LARC services, a number of initial investments are likely to be needed to catalyze the cycle of supply and demand for implants and IUDs. Both programs used donor funding to enable government-led training and mentorship and procure and distribute missing equipment needed for LARC service provision which resulted in rapid activation of facilities as access points for LARC services. Catalytic interventions, including traditional implementing partner supported mobile clinical outreach, can and should be linked to broader systems strengthening interventions, and where possible should seek to complement or help to facilitate commitment of subnational government resources to the implementation of program interventions. In both Uganda and Zambia, subnational government leadership have now integrated LARC clinical mentorship into SRMNH operational plans and budgets, with local governments making in-kind and financial commitments to the continuity of clinical mentorship. This suggests that subnational government structures and public health facilities may be able to take on more of the contraceptive-related program activities traditionally led by implementing partners and funded by international donors, integrating these to some extent into domestic budgets and plans.’’ 

To further expand upon recommendations we have added the following additional text: ‘’Strengthening government-owned supply chain systems to ensure availability of LARC commodities at facilities, while in tandem strengthening training and mentoring for health workers, is essential for enabling increased LARC service provision and use in the public sector. In parallel, barriers to demand must be meaningfully addressed, and further operational research to demonstrate cost-effective, sustainable, and scalable approaches to community mobilization and social and behavior change communication is needed to inform government and partner approaches as health system capacity increases. We recommend drawing from program learnings around the importance of using a systems lens to create a virtuous cycle of supply and demand when implementing aimed at increasing availability and voluntary uptake of LARCs and other family planning services in Uganda, Zambia, and comparable market contexts.’’

 

Reviewer #2 comments 

Summary: The authors conducted a study looking at strengthening systems to provide Long-Acting Reversible Contraceptives (LARCs) in public sector health facilities in Uganda and Zambia: Program results and learnings. This study is significant and applicable to both study settings as there is a high rate of unintended pregnancies. The article is generally well written, but the following points need to be considered:

1. The contextual differences or similarities between the two countries/study sites should be explained. This is crucial because it will enable a better understanding of the variations in project intervention uptake, such as the use of intrauterine devices (IUDs). The discussion's elaborated contextual explanations are insufficient. The authors might also consider other social and cultural aspects that have been covered in other works of literature.

Response: Thank you for this comment. Similarities in the health of the population were observed through review of maternal and newborn health indicators, with the chosen program regions scoring amongst the lowest comparatively. We note a number of important points of difference between the Uganda and Zambia contexts which may help explain the difference in results observed, including: ‘’Anecdotal reports from stakeholders indicate that in Uganda, previous IUD service provision by outreach teams without investment in public sector capacity to provide IUD side effect management and follow up care may have contributed to a cycle of low IUD provision and use. By contrast, in Zambia, health workers and users had limited experiences, both positive and negative, with IUDs. Strong opinions against the method amongst community members and key opinion leaders may have therefore been less fully formed.’’ We observed broad similarities across the common myths and misconceptions circulating in communities about LARCs, but found these were easier to shift in Zambia, in part likely due to the reasons stated above. We feel that a broader exploration of socio-cultural differences that impacted results may be beyond the scope of this paper, though agree that social-cultural enablers and barriers to family planning use are necessary to consider when designing holistic interventions to address both supply and demand side factors and have added an additional recommendation around the need for further operational research to demonstrate cost-effective, sustainable, and scalable approaches to community mobilization and social and behavior change communication (see response to question 21 above).

2. The authors ought to explain why particular provinces in the two countries were chosen. How does the provision of sexual, reproductive, maternal, and new-born health (SRMNH) in each nation compare on a provincial level?

Response: Thank you for this comment. We agree that more contextual information is useful and we have added the following text in the setting section in response:

“These areas of Zambia and Uganda were selected by the government as the focal area for this work because they were some of the most underperforming areas of the country on SRMNH-related outcomes, Northern Province, Zambia is a very rural and hard-to-reach areas with no other maternal and newborn health programs active in the province at the time that this program was launched. The selected districts in Uganda area are within several hours drive from the capital of Uganda, Kampala, but they nonetheless include areas that hard-to-reach.”

3. The authors must describe the type/category of health professionals trained at these facilities. Discuss how the project dealt with staff attrition throughout the course of the study as well.

Response: Thank you for this comment. We have added the following text in the program approach section to address this point:

“Training and mentorship was offered to cadres eligible to provide LARC services in each country, with a focus on nurses, midwives, and clinical officers. Facilities with trained HCWs received on-the-job mentoring. Mentorship helped to institutionalize skills within facilities and benefitted multiple health workers beyond those that were trained formally by the program. While mentoring may have helped to cascade skills and mitigate impacts of staff transfers, provider attrition was a reality the program had to contend with. The program coordinated with local leadership to rationalize staff transfers and ensure coverage of trained health workers at program sites. Trained health workers were encouraged to provide on-the-job training to other health workers in their facilities to ensure the availability of services in the event of attrition.”

Minor comments

4. Provide a reference for Figure 1. Virtuous cycle of LARC availability and provision.

Response: Thank you for requesting this clarification. The content of Figure 1 was developed by this program. We have added the following text to the paper to make that clear:

“Figure 1 depicts a virtuous cycle for LARC introduction as developed and used by this program.”

5. In the discussion section first paragraph “In Zambia, Marie Stopes Zambia was active in Zambian program geographies, and at the time of baseline data collection had recently started reporting into public sector facility registers.” Rephrase to avoid text redundancy

Response: Thank you for pointing out this opportunity to streamline. We have changed the text to the following:

“In Zambia, Marie Stopes was active in program geographies, and at the time of baseline data collection had recently started reporting into public sector facility registers.”

Journal Requirements:

Response: We have reviewed the style guidelines again and made several updates to ensure that the manuscript meets PLOS One’s guidance to best of our understanding. Please inform us if there are further changes needed.

Response: We have completed this questionnaire and uploaded it as requested.

Response: Thank you for your questions on consent. This work involved two types of data: from facility assessments and from aggregate routine service delivery data. Because no personal data was captured from or about individuals, both components were determined not to engage human subjects or to require consent. We have added the following text to the methods section to make this clear about each component:

Regarding facility assessments: “The IRBs determined that this work did not engage human subjects and that patient consent was not required.”

Regarding routine service delivery data: “This data is available in aggregate form without any individualized patient information, and as a result, this work was determined by CHAI’s internal Scientific and Ethical Review Committee to not engage human subjects or require patient consent.”

4. Thank you for stating in your financial disclosure: "This work was made possible with financial support from The ELMA Foundation and an anonymous donor (grants received and managed by HS and AM). The views expressed in this report are the opinions of the authors, and do not necessarily reflect the official policies of the funders. The funders had no role in study design, data collection and analysis, decision to publish, or preparation of the manuscript."

PLOS ONE requires you to include in your manuscript further information about the funder so that any relevant competing interests can be assessed. Please respond to the following questions:

a. Please state whether any of the research costs or authors' salaries were funded, in whole or in part, by a tobacco company (our policy on tobacco funding is at http://journals.plos.org/plosone/s/disclosure-of-funding-sources) 

b. Please state whether the donor has any competing interests in relation to this work (see http://journals.plos.org/plosone/s/competing-interests) . 

c. Please state whether the identity of the donor might be considered relevant to editors or reviewers’ assessment of the validity of the work.

d. If the donors have no perceived or actual competing interests, please state: “The authors are not aware of any competing interests”. 

This information should be included in your cover letter. We will amend your financial disclosure and competing interests on your behalf.

Response: We confirm the following:

a. No research costs or authors' salaries were funded, in whole or in part, by a tobacco company. 

b. The donor has no competing interests in relation to this work.

c. The donor had no role in study design, data collection and analysis, decision to publish, or preparation of the manuscript, and we do not feel that the identity of the donor would be considered relevant to editors or reviewers’ assessment of the validity of the work.

d. The authors are not aware of any competing interests.

We have added this information to our cover letter for the revision, as requested. Note that another publication emerging from this same body of work and with a similar financial disclosure statement was recently published in PLOS Global Public Health: 

Kamanga A, Ngosa L, Aladesanmi O, et al. Reducing maternal and neonatal mortality through integrated and sustainability-focused programming in Zambia. PLOS Glob Public Health. 2022;2(12):e0001162. Published 2022 Dec 14. doi:10.1371/journal.pgph.0001162

Please contact us if any further information is required regarding the financial disclosures.

5. Please ensure that you refer to Figures 2 to 5 in your text as, if accepted, production will need this reference to link the reader to the figure.

Response: Thank you for this point. We have included in text reference to these figures.

Response: We have updated the supplemental file references to the best of our understanding of the guidance available. Please inform us if there are further changes needed.

---

## [Editor Report · Decision Letter 1]

2 Aug 2023

Strengthening systems to provide Long-Acting Reversible Contraceptives (LARCs) in public sector health facilities in Uganda and Zambia: Program results and learnings

PONE-D-23-00410R1

Dear Dr. Aldrich,

We’re pleased to inform you that your manuscript has been judged scientifically suitable for publication and will be formally accepted for publication once it meets all outstanding technical requirements.

Kind regards,

Lea Sacca

Academic Editor

PLOS ONE
---

## [Editor Report · Acceptance letter]

11 Aug 2023

PONE-D-23-00410R1 

Strengthening systems to provide Long-Acting Reversible Contraceptives (LARCs) in public sector health facilities in Uganda and Zambia: Program results and learnings 

Dear Dr. Aldrich:

I'm pleased to inform you that your manuscript has been deemed suitable for publication in PLOS ONE. Congratulations! Your manuscript is now with our production department. 

Kind regards, 

on behalf of

Dr. Lea Sacca 

Academic Editor

PLOS ONE